# Theory of universal diode effect in three-terminal Josephson junctions

J. H. Correa[1†] and M. P. Nowak[1*]

**1** AGH University of Krakow, Academic Centre for Materials and Nanotechnology, al. A. Mickiewicza 30, 30-059 Krakow, Poland

† jorge@agh.edu.pl * mpnowak@agh.edu.pl

January 30, 2024

## Abstract

We theoretically study the superconducting diode effect in a three-terminal Josephson junction. The diode effect in superconducting systems is typically related to the presence of a difference in the critical currents for currents flowing in the opposite direction. We show that in multi-terminal systems this effect occurs naturally without the need of the presence of any spin interactions and is a result of the presence of a relative shift between the Andreev bound states carrying the supercurrent. On an example of a three-terminal junction, we demonstrate that the non-reciprocal current in one of the superconducting contacts can be induced by proper phase biasing of the other contacts, provided that there are at least two Andreev bound states in the system and the symmetry of the system is broken. This result is confirmed in numerical models describing the junctions in both the short- and long-regime. By optimizing the geometry of the junction, we show that the efficiency of the realized superconducting diode exceeds 35%. We relate our predictions to recent experiments on multi-terminal junctions, in which non-reciprocal supercurrents were observed.

# 1 Introduction

In recent years the use of superconductors to create electronic elements that show non-reciprocal behavior—superconducting diode effect (SDE)—has attracted great interest [1–5]. The realization of this phenomenon in systems consisting of Josephson junctions (JJs) coined the name Josephson diodes for such devices. The diode effect has recently been investigated in several experimental and theoretical works that considered graphene JJ [6], Andreev molecules [7–9], artificial superlattices [10], twisted materials [11–13], van der Waals heterostructures [14], topological semimetals and insulators [15–17], insulator heterostructure devices [18], nanowires [19], transition lines [20], 3D nanobrigdes [21], and disordered systems [22]. The possibility of obtaining non-reciprocal behavior in these junctions also shows a profound impact on the creation of electronic devices such as photodetectors, transistors, ac/dc converters, superconducting qubits, and devices that exhibit Shapiro steps [23, 24].

In a single JJ the appearance of the non-reciprocal current and SDE can be induced by breaking the time-reversal symmetry and the inversion symmetry [4,16], which can be achieved by the action of an external magnetic field through the Zeeman effect and the Rashba spin-orbit coupling (RSOC) [19, 25, 26]. However, it has been found that SDE can also be obtained by breaking either of the two symmetries separately [3], as well as other symmetries [27]. These broken symmetries lead to a shift in ABSs energies and the appearance of higher harmonic terms in the current phase relation (CPR) [19] analogously to the case of two JJs in a SQUID loop, leading to the difference between maximum and minimum critical supercurrents [28].

An alternative route for the realization of the SDE is to use multi-terminal systems. The Josephson diode effect was, in fact, already observed in multi-terminal JJs. Chiles et al. [6] showed that in a system with three graphene JJs commonly linked to a superconducting island, it is possible to achieve rectification of the supercurrent without the need for an external magnetic field by applying a dissipation-less control current at one of the junctions. However, the unique feature of multi-terminal systems is that they allow for alteration of the ABS spectrum and the resulting CPR by proper phase biasing [29]. Phase biasing has already been experimentally exploited to induce the diode effect in coupled JJs [9] in or in a 2DEG connected to multiple superconducting leads [30, 31]. However, those works considered that the systems consist of a few *separate* JJs, either in an Andreev molecule configuration [7] or with the junctions connected in parallel. Here, we explore a fundamental process that underlies the SDE in multi-terminal systems. Using both analytical and numerical models, we show that the diode effect can naturally emerge in a *single* multi-terminal junction (single-scattering region, multiple superconducting leads) due to the presence of several ABSs that couple to the phase-biased superconducting leads with different magnitudes and hence experience a relative phase shift.

The outline of this paper is as follows. In Sec. 2 we present a proof-of-concept model with an analysis of the short junction regime pointing out the origins of non-reciprocal currents. Then, we perform numerical calculations to support our analytical findings and demonstrate the SDE in a three-terminal JJ. In Sec. 3, we extend our investigation of the SDE beyond the short junction regime. Finally, in Section 4, we provide a discussion and conclusions.

# 2 Short junction regime

## 2.1 Proof-of-concept model

Let us consider a three-terminal JJ shown schematically in Fig. 1 where three superconducting leads (with corresponding pairing potentials $\Delta e^{i\phi_i}$) are connected through a common normal

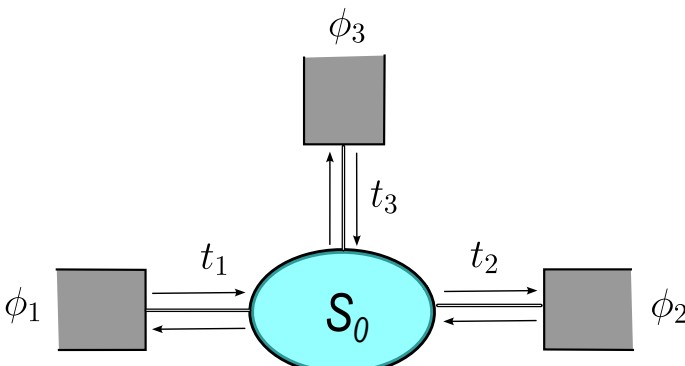

Figure 1: Schematic picture of a three-terminal JJ realized by three superconducting leads (grey) connected through a central scatterer (blue). The strength of the coupling of the superconducting terminals to the central scattering region is denoted with $t_i$.

scattering region. In general, the ABS energy in a multi-terminal junction is given by [24, 32]

$$\frac{E}{\Delta} = \pm\frac{1}{2}\sqrt{1 + Tr(S\mathbf{e}^{i\phi}S^*\mathbf{e}^{-i\phi})}, \tag{1}$$

with $\mathbf{e}^{\pm i\phi}$ a diagonal matrix consisting of the phases in each lead and with $S$ the scattering matrix of the region between the superconductors.

Using Eq. 1, the energies are given by

$$E = \pm\frac{\Delta}{2}\sqrt{1 + \sum_{l,j} T_{lj} e^{-i(\phi_l - \phi_j)}}, \tag{2}$$

where $T_{lj}$ are the transmission probabilities for the quasiparticle injected in the $l$'th lead to reach the $j$'th terminal obtained from the elements of the scattering matrix $S$ as $T_{lj} = |s_{lj}|^2$. When the time-reversal symmetry of the *scattering region* is preserved the scattering matrix is symmetric $S = S^T$ and the formula for ABS energy for a three-terminal JJ can be simplified further into $E = \pm\Delta\Pi$ with [24, 33]

$$\Pi = \left[1 - T_{21}\sin^2\left(\frac{\phi_2 - \phi_1}{2}\right) - T_{31}\sin^2\left(\frac{\phi_3 - \phi_1}{2}\right) - T_{23}\sin^2\left(\frac{\phi_2 - \phi_3}{2}\right)\right]^{1/2}. \tag{3}$$

The supercurrent in the $l$'th superconducting lead is obtained from positive energy ABS as [34–36]

$$I_l = -\frac{2e}{\hbar}\tanh\left(\frac{E}{2k_bT}\right)\frac{dE}{d\phi_l}, \tag{4}$$

with the 2 factor accounting for the spin degeneracy.

Let us focus on the current in the second terminal ($l = 2$), while we allow for an arbitrary phase bias in the third. Therefore, using Eqs. 2, 3 and 4 and at zero temperature, we can express the current as

$$I_2 = \frac{e\Delta}{2\hbar\Pi}\left[T_{21}\sin(\phi_2 - \phi_1) + T_{23}\sin(\phi_2 - \phi_3)\right]. \tag{5}$$

This result indicates a non-local behavior where the current flowing in one lead is influenced by the phase difference between the other leads. In the following, we set the gauge $\phi_1 = 0$ that sets the reference point for the phase differences between the superconducting terminals.

For the analysis of the ABS spectrum and currents, we need to further establish the scattering matrix of the normal region and, therefore, the transmission coefficients that appear in the Eqs. 3 and 5. We first assume that the normal region between the superconductors is an ideal, single-mode beam splitter whose scattering matrix $S_0$ is [24],

$$S_0 = \begin{pmatrix} r & \tau & \tau \\ \tau & r & \tau \\ \tau & \tau & r \end{pmatrix}, \tag{6}$$

where $\tau$ and $r$ are the transmission and reflection coefficients, respectively, equal to $\tau = 2/3$ and $r = -1/3$ for the perfectly transparent splitter.

The quasiparticle transport properties between the superconducting contacts can be contained in the complete scattering matrix of the system $S$, which we write as [37],

$$S = S_{PP} + S_{PQ} S_0 \frac{1}{I - S_{QQ} S_0} S_{QP}, \tag{7}$$

The matrices that describe the coupling of the center scattering region to the superconducting leads are defined as follows

$$S_{PP} = \begin{pmatrix} r_1' & 0 & 0 \\ 0 & r_2' & 0 \\ 0 & 0 & r_3' \end{pmatrix}, \quad S_{PQ} = \begin{pmatrix} t_1 & 0 & 0 \\ 0 & t_2 & 0 \\ 0 & 0 & t_3 \end{pmatrix}, \tag{8}$$

and

$$S_{QP} = \begin{pmatrix} t_1' & 0 & 0 \\ 0 & t_2' & 0 \\ 0 & 0 & t_3' \end{pmatrix}, \quad S_{QQ} = \begin{pmatrix} r_1 & 0 & 0 \\ 0 & r_2 & 0 \\ 0 & 0 & r_3 \end{pmatrix}. \tag{9}$$

$t_i$ are the coupling amplitudes between the $i$'th superconductor and the normal region, and $r_i = \sqrt{1 - t_i^2}$ are the reflection amplitudes [see Fig. 1]. The primed values correspond to the amplitudes of a time-reversed transport process with $t_i' = t_i$ and $r_i' = -r_i$. The coefficients $s_{12}$, $s_{13}$ and $s_{23}$ used to obtain the transmission probabilities in Eq. 3 are given by:

$$s_{12} = \frac{2t_1 t_2 (-r_3 - 1)}{B}, \qquad s_{13} = \frac{2t_1 t_3 (-r_2 - 1)}{B}, \qquad s_{23} = \frac{2t_2 t_3 (-r_1 - 1)}{B}, \tag{10}$$

where $B = 3r_1 r_2 r_3 + r_1 r_2 + r_1 r_3 + r_2 r_3 - r_1 - r_2 - r_3 - 3$.

Let us consider a minimal case in which the diode effect can be realized in our system—when the ABSs spectrum consists of two states, each described by Eq. 3. First, we decouple the third superconducting lead by setting $t_3 = 0$. It is clear that in this case, Eq. 3 reduces to the known formula $E = \pm\Delta\sqrt{1 - T_{12}\sin^2(\Delta\phi/2)}$, where $\Delta\phi = \phi_2 - \phi_1$ and where $T_{12}$ is the transmission coefficient between the two leads [38], with fully reciprocal behavior, that is, $I_c^+(\Delta\phi) = -|I_c^-(\Delta\phi)|$.

Now, let us consider a finite coupling to the third superconducting terminal that we bias by phase $\phi_3 = 1.5\pi$. We consider the situation where $t_1 = t_2 \neq t_3$, which effectively breaks the previously introduced perfect symmetry of the beam splitter. In Fig. 2(a) we observe that the energy-phase relations of the two ABSs are shifted with respect to each other due to the different strengths of the coupling to the third terminal. In Fig. 2(b) we show the corresponding supercurrents calculated using Eq. 5 for these two ABSs. The current carried by the state strongly coupled to the third terminal is non-symmetric with respect to $\phi_2 = \pi$ (green curve), and its phase-shift depends on the value of $\phi_3$. The presence of the third superconducting lead induces an anomalous current [19, 39] carried by one of the ABSs, but

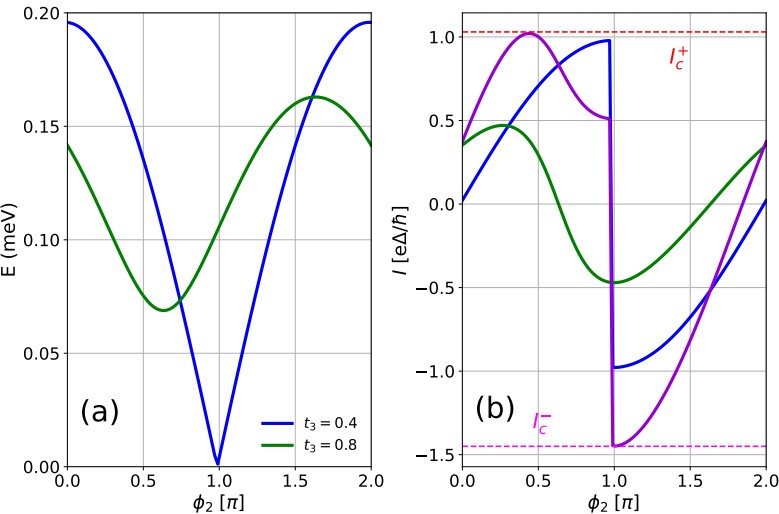

Figure 2: Energy spectrum (a) and supercurrent (b) in three-terminal JJ hosting two ABSs. $t_1 = t_2 = 1$ (blue curve) and, $t_1 = t_2 = 0.6$ (green curve), the violet curve in (b) shows the non-reciprocal current carried by the two ABSs. The phase on the third superconducting terminal is $\phi_3 = 1.5\pi$.

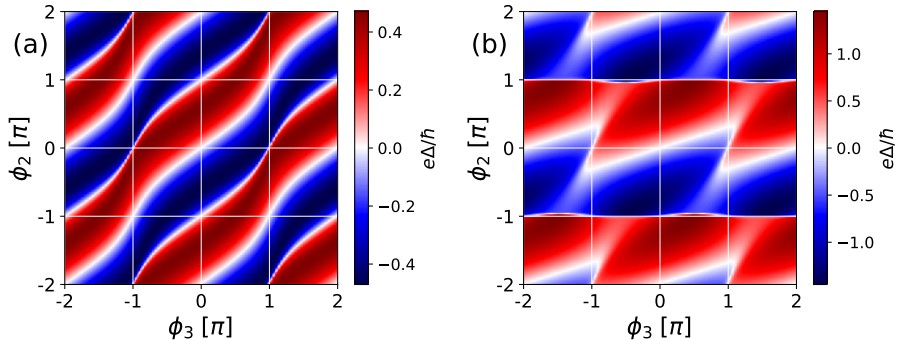

Figure 3: Map of supercurrent flowing in lead 2 as a function of $\phi_2$ and $\phi_3$ for single ABS with $t_1 = t_2 = 0.6$ and $t_3 = 0.8$ (a) and for two modes (b) for the same parameters of $t_1$, $t_2$ and $t_3$ as in Fig. 2.

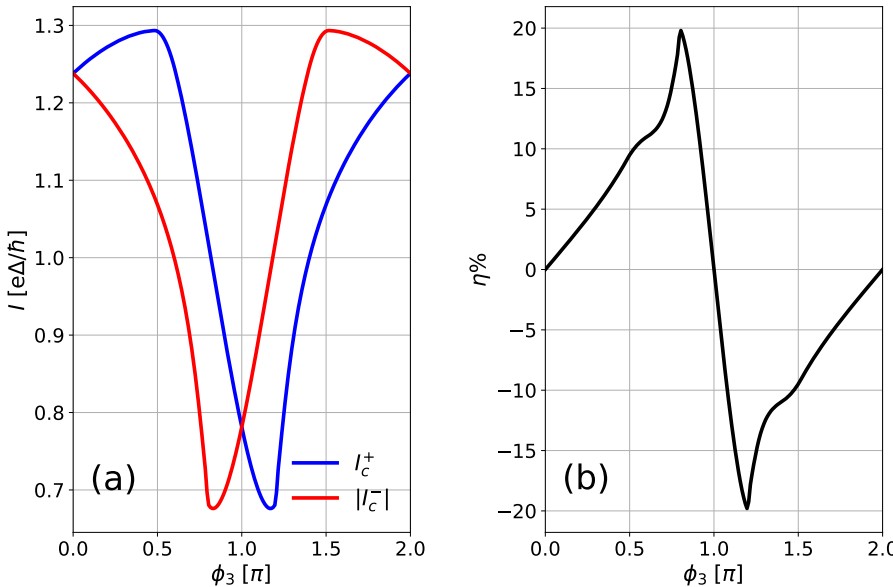

Figure 4: (a) Minimal and maximal values of the critical current versus the phase on the third superconducting lead. (b) SDE efficiency. The results are obtained for 50 modes with $t_1 = t_2 = 0.8$ and a $t_3$ values chosen uniformly in the range $[0,1]$.

each mode separately conducts a reciprocal current. However, the phase shifts between the currents carried by each ABS introduce an amplification of the supercurrent in part of the phase range, while a decrease of the supercurrent is found in the other, which in turn results in the SDE as can be seen in Fig. 2(b). It should also be noted that the phase shift of the ABS is controlled not only by the strength of the coupling to the third terminal ($t_3$) but also by the probabilities $t_1$ and $t_2$ that affect the $T_{ij}$ coefficients that stand next to the $\phi_3$-dependent terms in Eq. 3.

On the map of Fig. 3(a), we plot the current carried by the ABS shifted in phase (green in Fig. 2(a)). The map signifies both local inversion symmetry breaking and local time inversion symmetry breaking, that is, $I(\phi_2, \phi_3) \neq -I(-\phi_2, \phi_3)$, however, this symmetry breaking does not lead to an SDE and also that the global time-reversal symmetry is preserved $I(\phi_2, \phi_3) = -I(-\phi_2, -\phi_3)$ [8,9,31]. On the other hand, in Fig. 3(b), we show the supercurrent map obtained for the two modes. Here also the inversion symmetry is broken and the global time-reversal symmetry is preserved, but in contrast to panel (a), for the non-zero value of $\phi_3$, in each $\phi_2$ current cross-section its minimum and maximum are different giving rise to SDE, as shown in the violet curve of Fig. 2(b) with $I_c^+(\phi_2) \neq -|I_c^-(\phi_2)|$.

The strength of SDE can be characterized by its efficiency, which we define as

$$\eta = \frac{I_c^+ - |I_c^-|}{I_c^+ + |I_c^-|}. \tag{11}$$

It is clear that for a single mode $\eta = 0$, and for the parameters that we consider for two modes, we obtain $\eta \approx -17\%$, a larger critical current flowing in the negative direction than in the positive one.

Although, as we have shown, it is possible to achieve SDE already in the presence of two ABSs, in general, the system can consist of a much larger number of states carrying the supercurrent. Therefore, we consider the minimum and maximum supercurrents and the corresponding SDE efficiency obtained for a system with 50 ABSs with a uniform distribution of $t_3$ in the $[0, 1]$ range. In Fig. 4 we show $I_c^+$ and $|I_c^-|$ as a function of $\phi_3$ (a) and the corresponding

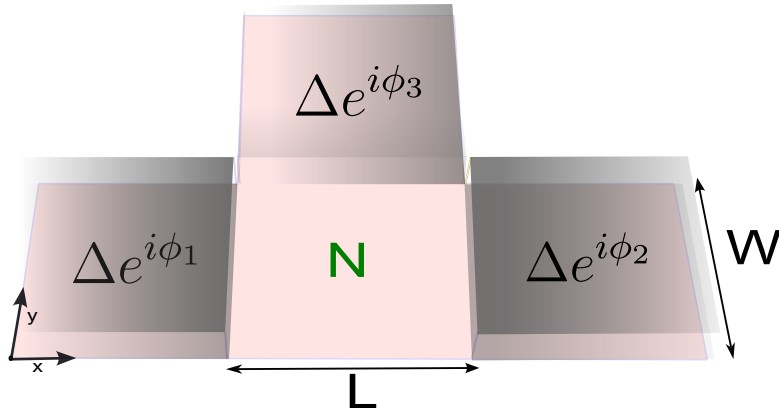

Figure 5: Planar JJ considered in the numerical calculations. The pink region corresponds to the normal part of the junction, and the gray superconducting segments are the superconducting leads.

SDE efficiency (b). We see that the curves show mirror symmetry with respect to $\phi_3 = \pi$ and the system reaches high-efficiency values around the vicinity of this point, however, the efficiency goes zero for $\phi_3 = 0 \pmod{\pi}$, which indicates that the inversion symmetry is preserved at these points.

With our analytical approach, we have determined that the SDE is an intrinsic property in the three-terminal JJ with the requirement of the junction embedding more than one ABS with different coupling to the phase-biased terminal. We show that the current contributions resemble the non-harmonic components of the current previously considered in a two-terminal JJ, where this shift is attributed to the finite momentum of the Cooper pairs due to the action of an external magnetic field [2, 15, 19].

## 2.2  Numerical model—short junction approximation

In general, the SNS junction can host an arbitrary number of ABSs and its spectrum can be determined from the matching condition $S_A(E) S_N(E) \Psi_{in} = \Psi_{in}$, where $\Psi_{in} = (\Psi_e, \Psi_h)$ are the complex amplitudes of the electron and hole waves incident at the junction defined in the basis of normal region scattering modes. $S_A(E)$ describes an Andreev reflection process at the NS interface which in the case of the absence of the mode mixing at the interface is [29, 40–42]

$$S_A(E) = \zeta(E) \begin{pmatrix} 0 & r_A^* \\ r_A & 0 \end{pmatrix}, \tag{12}$$

with the amplitude $\zeta(E) = \sqrt{1 - E^2/\Delta^2} + iE/\Delta$ and $r_A$ the Andreev reflection matrix, whose dimension depends on the number of the superconducting leads [29]. Assuming that the outgoing modes are time-reversed equivalents of the incoming modes, for a three-terminal junction we have

$$r_A = \begin{pmatrix} ie^{i\phi_1} \mathbf{1}_{n_1} & 0 & 0 \\ 0 & ie^{i\phi_2} \mathbf{1}_{n_2} & 0 \\ 0 & 0 & ie^{i\phi_3} \mathbf{1}_{n_3} \end{pmatrix}. \tag{13}$$

The block-diagonal matrix that captures the scattering properties for electrons and holes in the normal region is given by

$$S_N(E) = \begin{pmatrix} S(E) & 0 \\ 0 & S^*(-E) \end{pmatrix}, \tag{14}$$

with $S(E)$ ($S^*(-E)$) the electron (hole) scattering block.

We assume a short-junction regime with the superconducting coherence length $\xi = v_f \hbar / \Delta$ much larger than the dimensions of the normal region which allows us to simplify $s \equiv S(E = 0)$, and to arrive at the eigenvalue problem

$$\begin{pmatrix} s^\dagger & 0 \\ 0 & s^T \end{pmatrix} \begin{pmatrix} 0 & r_A^* \\ r_A & 0 \end{pmatrix} \Psi_n = \zeta(E)\Psi_n, \tag{15}$$

whose solution yields the set of ABS eigenenergies and wave functions.

We use the above-mentioned model to simulate a planar three-terminal junction. We consider a normal region attached to three semi-infinite superconductor leads, forming a $T$-shaped junction, which is schematically depicted in Fig. 5. We assume a typical Hamiltonian for a semiconducting normal region

$$H_N = \left( \frac{\hbar^2 \mathbf{k}^2}{2m^*} - \mu \right) \sigma_0 + \alpha(\sigma_x k_y - \sigma_y k_x). \tag{16}$$

This comprises both the kinetic energy part for the charge carriers and the Rashba spin-orbit coupling (RSOC) contribution. $m^*$ is the effective electron mass, $\mu$ is the chemical potential, $\alpha$ controls the spin-orbit coupling strength, and $\sigma = (\sigma_0, \sigma_x, \sigma_y, \sigma_z)$ are the Pauli matrices. We consider a ballistic case, that is, when the electron mean free path $l_e$ is much larger than the dimensions of our system.

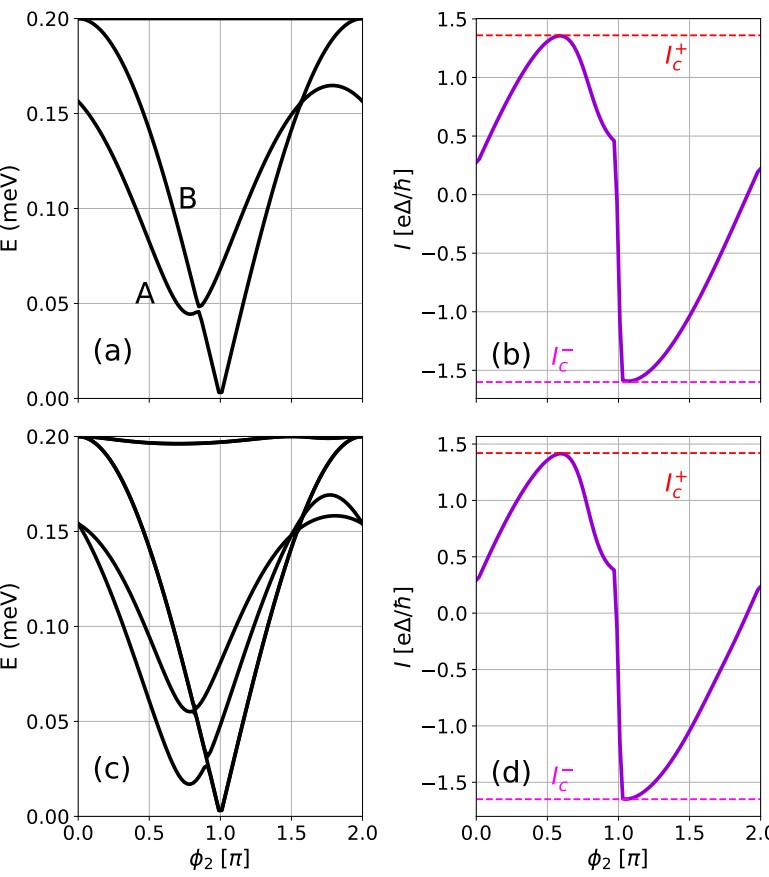

Figure 6: ABS energies (a), (c) and the supercurrent (b), (d) obtained numerically for a short-junction for $L = 50$ nm and $W = 120$ nm, $\mu = 10$ meV, $\phi_3 = 1.5\pi$. (a) and (b) are obtained with $\alpha = 0$ while (c) and (d) for $\alpha = 50$ meVnm.

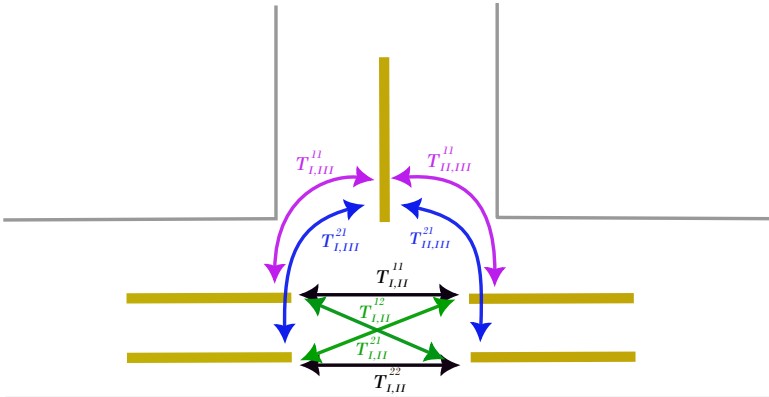

Figure 7: Schematic plot of the transmission probabilities (color arrows) between the modes (yellow) in the leads attached to the scattering region.

For concreteness, we set the effective mass corresponding to the semiconductor commonly used in hybrid junctions, namely InSb [43, 44] with the effective mass $m = 0.014m_e$ and the superconducting gap corresponding to that of aluminum $\Delta = 0.2$ meV. Although this is an arbitrary choice, it does not affect the generality of the phenomena, which we discuss here. We discretize the Hamiltonian Eq. 16 on a square lattice with the lattice constant $a = 5$ nm and obtain the scattering matrix using Kwant package [45]. We set $\phi_1 = 0$ and the supercurrent in the second terminal at zero temperature is calculated from the ABS spectrum $E_n(\phi_2, \phi_3)$ analogously to Eq. 4 by including the contribution of all positive-energy ABS: $I_2(\phi_2) = -e/\hbar \sum_{E_n>0} dE_n/d\phi_2$.

Let us first consider a spin-degenerate case neglecting the RSOC ($\alpha = 0$ in Eq. 16). In Fig. 6 we show the ABS energies (a) and supercurrent (b), considering $L = 50$ nm, $W = 120$ nm, $\mu = 10$ meV, and $\phi_3 = 1.5\pi$. It is important to note here that the number of ABS obtained through Eq. 15 crucially depends on the number of charge-carrying bands in each normal lead connected to the superconducting terminal. The dimension of the scattering matrix $s$ is $(N, N)$, where $N$ is the sum of the number of modes in all leads. As a result, the number of ABS is $\lceil N/2 \rceil$. For the case of Fig. 6(a) we have $N = 5$ spin-degenerate modes, and there are two current-carrying ABSs and one which is located at the gap energy.

The two phase-dependent ABSs presented in Fig. 6(a) show a behavior similar to the ones obtained in the analytical model in Fig. 2(a). Positive and negative critical current values are indicated by dashed lines in Fig. 6(b). As we can see, the current flowing in the negative direction is slightly different from the positive one, generating the SDE, and the system reaches an efficiency of $-8\%$. Here, the natural question arises whether the ABS structure and the SDE effect here have the same origin as in the proof-of-concept model case.

In Table 1, we denote by $T_{I,J}^{K,L}$ the transmission probability between the $K$'th mode in the $I$'th lead and the $L$'th mode in the $J$'th lead. In the normal part of the system, where the time-reversal symmetry is preserved, we have $T_{I,J}^{K,L} = T_{I,J}^{L,K}$ and also $T_{I,J}^{K,L} = T_{J,I}^{K,L}$. In Fig. 7, we schematically denote the transmission elements that connect the modes between the three leads. It is clear that even though our system is ballistic, the transmission probability between the horizontal leads is different from the one between one of the horizontal and vertical leads. This is a clear effect of the broken $C3$ symmetry of the considered system.

For each ABS, which are the solution of Eq. 15 its eigenvector components consist of the amplitudes of the normal-state wave-functions obtained from incoming electron and holes from the corresponding superconducting leads. In Table 2 we show the absolute square elements of the $\Psi$ eigenvectors for the two phase-dependent ABS shown in Fig. 6(a) for $\phi_2 = 0.5\pi$ denoted with $A$ and $B$ in Fig. 6(a). It is striking that the ABS that is shifted in phase ($A$) is

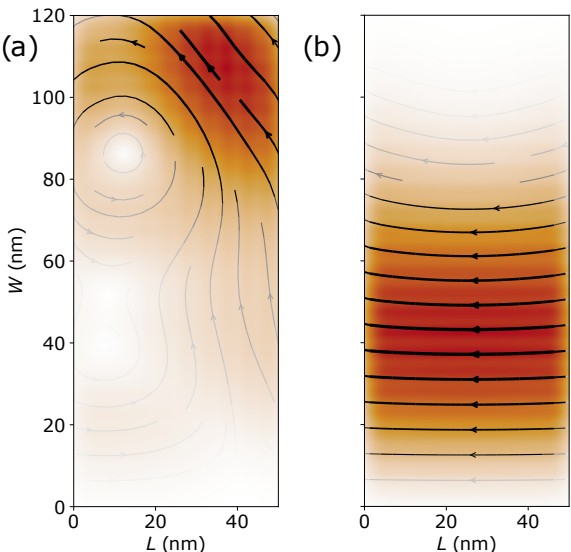

Figure 8: Supercurrent densities for two ABSs denoted with $A$ (a) and $B$ (b) in Fig. 6(a) obtained for $\phi_2 = 0.5\pi$.

| $T_{I,II}^{1,1}$ | $T_{I,II}^{2,2}$ | $T_{I,II}^{1,2}$ | $T_{I,III}^{1,1}$ | $T_{I,III}^{2,1}$ | $T_{II,III}^{1,1}$ | $T_{II,III}^{2,1}$ |
|---|---|---|---|---|---|---|
| 0.93 | 0.534 | 0.016 | 0.043 | 0.346 | 0.043 | 0.346 |

Table 1: Transmission probabilities between the normal leads for subsequent modes of transverse quantization for $L = 50$ nm, and $W = 120$ nm.

constructed from virtually all modes, while the second one ($B$) is constructed mostly from the modes incoming from the left and right leads.

We find that ABS $A$ is characterized by a considerable coupling of the upper terminal with the horizontal one, provided by a significant amplitude of the incoming modes from the top contact and a considerable value of $T_{I,III}^{1,1} = T_{II,III}^{1,1} = 0.043$ and $T_{I,III}^{2,1} = T_{II,III}^{2,1} = 0.346$. For ABS $B$, the situation is the opposite. This ABS does not include quasiparticles coming from the top terminal. However, it is characterized by a strong coupling between the left and right terminals because of the dominating contribution of the quasiparticle modes entering from them and large $T_{I,II}^{1,1}$ and $T_{I,II}^{2,2}$ values.

This finding is further confirmed by the supercurrent distribution carried by each ABS shown in Fig. 8, where we observe that ABS $A$ (panel(a)) carries the current mainly between the right and the top lead, while ABS $B$ (panel(b)) carries the current only between the left and right contacts. We then see that the considered junction, due to broken symmetry of the structure, the scattering probability between horizontal and vertical leads is different and the

| ABS no. | $I,1$ | $I,2$ | $II,1$ | $II,2$ | $III,1$ |
|---|---|---|---|---|---|
| A electron | 0.01 | 0.087 | 0.04 | 0.316 | 0.046 |
| B electron | 0 | 0 | 0.445 | 0.055 | 0 |
| A hole | 0.02 | 0.149 | 0.003 | 0.024 | 0.305 |
| B hole | 0.444 | 0.056 | 0 | 0 | 0 |

Table 2: Squared modulus of the electron and hole elements of the ABSs eigenvectors. The subsequent columns correspond to the amplitudes for the wave-functions obtained for the electron/hole injected in $I$, $II$ or $III$ lead in the first or the second mode.

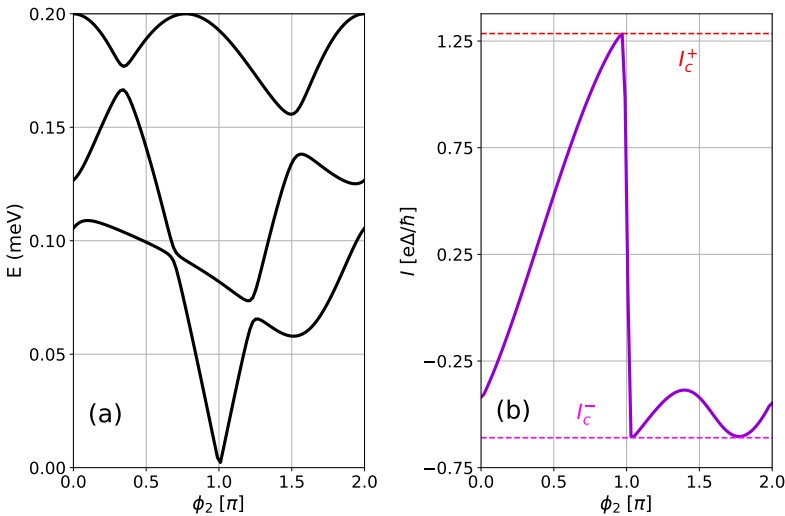

Figure 9: ABS energies (a) and supercurrent (b) in three-terminal JJ as a function of $\phi_2$ with $L$, $W$ and $\phi_3$ tuned for the maximal efficiency. The optimized values are $L = 100$ nm, $W = 125$ nm and $\phi_3 \approx 0.77\pi$.

presence of a few scattering modes in the wider contacts leads to the creation of two ABSs that are characterized with different capabilities for transporting quasiparticles between horizontal and vertical leads, resulting in the SDE effect for non-zero phase bias $\phi_3$ in accordance with our proof-of-concept model.

Finally, we include in our calculation the RSOC that is usually strong in III-V 2DEGs or nanowires used for the creation of multi-terminal junctions [1,10,25,46] and inspect its impact on the diode effect. In Figs. 6(c) and (d) we show the ABS spectrum and supercurrent of the same system as considered before but taking into account $\alpha = 50$ meVnm. As we see, the main effect of this term is to break the spin degeneracy in the ABSs spectrum for the ABSs affected by $\phi_3$, however, the supercurrent Fig. 6(d) does not undergo significant changes and the system still reaches $\eta \approx -7\%$.

### 2.2.1 Efficiency optimization

As we demonstrated, the SDE appears already in a few-mode JJ. In this section, we analyze the geometrical properties of the junction that can be altered to obtain the highest efficiency. Using numerical optimization of efficiency over the parameters $L$, $W$, and $\phi_3$ we find the spectrum and current presented in Figs. 9(a) and (b), respectively. In this case, the optimal values are $L = 100$ nm, $W = 125$ nm, and $\phi_3 \approx 0.77\pi$, and the system reaches the efficiency of $\eta \approx 36\%$. We checked and found that the inclusion of RSOC gives practically the same efficiency values.

In the map of Fig. 10 we present the optimized efficiency values versus the length and width of the normal region. First, on the map a clear grid-like structure is observed. It results from the rapid jump in efficiency when $W$ or $L$ increases so that the subsequent transverse quantization mode enters below the Fermi energy. We observe that the highest efficiencies are obtained mostly for uniform systems, that is, when the length and width are comparable, resulting in the same number of modes in the horizontal and vertical leads. Finally, we see that for $W$ and $L$ less than 100 nm, the efficiency is zero, since there is only one mode in each lead, which leads to a single ABS carrying the current, and therefore the lack of the SDE effect, as discussed before.

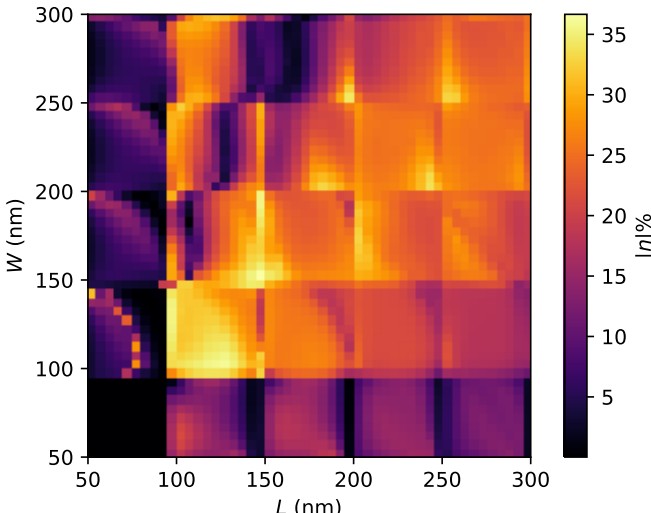

Figure 10: Map of the absolute values of SDE efficiency ($\eta$) as a function of length ($L$) and width ($W$) of the normal region without RSOC, considering the optimized value of $\phi_3$ for each point of $(L, W)$.

## 3 Beyond the short junction regime

So far we have focused on the case of short junction, where: i) the energy dependence of the scattering events was not taken into account; ii) only the subgap ABSs were taken into account when calculating the current. Here, we consider a situation where the superconducting coherence length $\xi$ is smaller than the dimensions of the normal region, which is often the case especially when using superconductors with a large gap such as $TaN_x$ [47], $Nb$ [48], $MgB_2$ [49].

We investigate the long-junction regime, adopting a model that goes beyond the limitations of the short-junction model mentioned above. We describe the whole SNS junction by the Hamiltonian

$$H = \begin{pmatrix} H_N & \Delta(x,y)\sigma_0 \\ \Delta^*(x,y)\sigma_0 & -H_N \end{pmatrix}, \tag{17}$$

with the superconductor pairing potential defined as

$$\Delta(x,y) = \begin{cases} \Delta & \text{if } -L_{SC} < x < -L/2 \\ 0 & \text{if } -L/2 \leq x \leq L/2 \\ \Delta e^{i\phi_2} & \text{if } L/2 < x < L_{SC} \end{cases}, \tag{18}$$

with $0 < y < W$ and

$$\Delta(x,y) = \Delta e^{i\phi_3}, \quad -L/2 \leq x \leq L/2 \text{ and } L_{SC} > y > W. \tag{19}$$

The above model essentially leads to the same system as described previously but with infinite leads replaced by finite superconducting segments of length $L_{sc} \gg \xi$. We take $\Delta = 1$ meV, which results in $\xi \approx 330$ nm. We assume that the normal system is symmetric with $L = W = 500$ nm and take $L_{sc} = 1000$ nm. We obtain the energy spectrum of the junction by diagonalizing the Hamiltonian Eq. 17 discretized on a square lattice with lattice constant $a = 10$ nm and subsequently calculate the current by differentiating the positive part of the energy spectrum in the same manner as in the short-junction case.

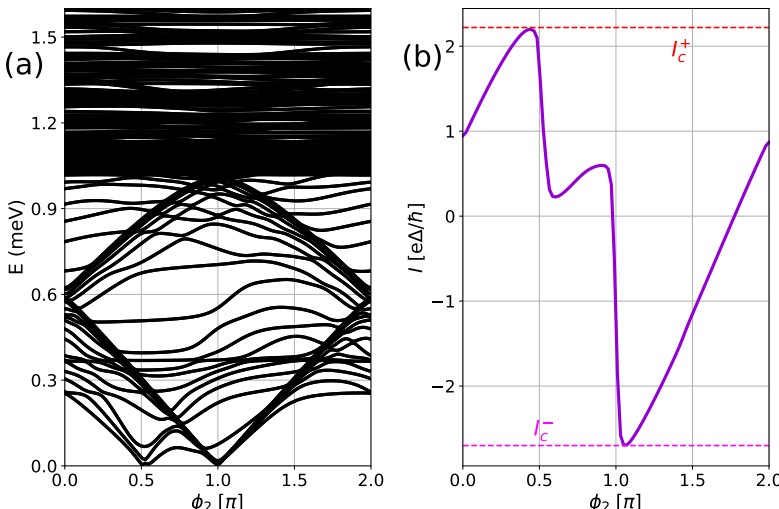

Figure 11: ABS energies (left panel) and supercurrent (right panel) in three terminal JJ in long junction regime as a function of $\phi_2$. $L = W = 500$ nm and $\phi_3 = 1.5\pi$.

It is clear that Hamiltonian given by Eq. 17 preserves the global time-reversal symmetry that is $\mathcal{T}H(\phi_2, \phi_3)\mathcal{T}^{-1} = H(-\phi_2, -\phi_3)$ being $\mathcal{T}$ the antiunitary complex conjugate, nevertheless this symmetry is broken locally $\phi_2 \to -\phi_2$ for fixed $\phi_3 \neq 0$ (mod $\pi$), likewise the spacial inversion symmetry is also broken locally $\mathcal{I}H(\phi_2, \phi_3)\mathcal{I}^{-1} \neq H(-\phi_2, \phi_3)$ with $\mathcal{I}$ the unitary inversion operator.

In Fig. 11, we present the ABS energy spectrum and supercurrent distribution in the (a) and (b) panels, respectively, considering $\phi_3 = 1.5\pi$. It is important to note here that the spectrum consists of phase-dependent states with energies above the superconducting gap that form a quasi-continuum and have to be taken into account when calculating the supercurrent [2, 8]. Below the energy of the superconducting gap we observe many ABSs that can be grouped into two families: those with the minima located at $\phi_2 = \pi$ and those that are visibly shifted in phase, with the minimum located at $\phi_2 \simeq 0.5\pi$. The relative shift between the two types of ABS again leads to the SDE effect as presented in the supercurrent plot in Fig. 11(b) despite the overall change in the character of the ABSs structure due to the extended length of the junction and the large superconducting gap value. We observe that the values of the maximum and minimum critical currents differ significantly from each other, resulting in the efficiency of $-10\%$. Moreover, $I(\phi_2 = 0) \neq 0$, leading to an anomalous current.

In Fig. 12 (a), we show the maximum and absolute minimum critical supercurrent values, where the asymmetry is observed in all phase values except $\phi_3 = 0$ (mod $\pi$). Furthermore, in Fig. 12 (b), we have plotted the efficiency as a function of $\phi_3$. We observe that the system reaches high values in the center of the plot and goes to zero for $\phi_3 = 0$ (mod $\pi$) where the inversion symmetry is preserved. We also observe that by changing the sign of the $\phi_3$ sign we can reverse the polarity of the diode.

# 4 Discussion and Conclusions

In this work, we have analyzed the physical origin of the superconducting diode effect in multi-terminal Josephson junctions based on the example of a three-terminal device. We have demonstrated both analytically and numerically that the diode effect can naturally occur in a few-mode SNS junction, provided that the junction is biased by the superconducting phase at one of the terminals to which the ABSs states present in the junction couple with different

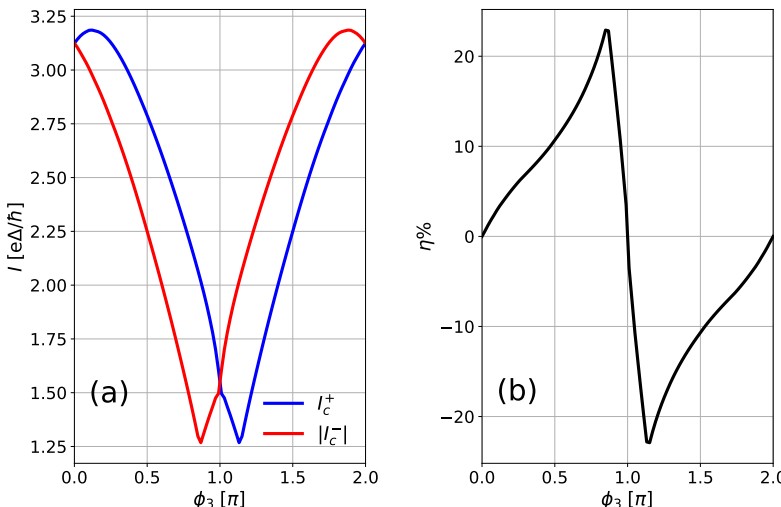

Figure 12: Maximum ($I_c^+$) and the absolute value of minimum ($|I_c^-|$) values of the critical current (a) and efficiency (b) as a function of $\phi_3$ for a long JJ with $L = W = 500$ nm and $\Delta = 1$ meV.

magnitudes, which in turn can occur in a multimode three-terminal system with broken $C3$ symmetry.

Phase biasing, exploited in this work for the realization of the SDE can be experimentally achieved by connecting two terminals of the junction into a large superconducting loop [50–52] and threading such a loop with a small magnetic flux. In fact, such a three-terminal configuration was recently realized in a Josephson molecule system [9]. There, the three-terminal system consisted of a single superconducting lead connected by two normal regions to two outer superconducting contacts. This two-Josephson junction system acts as a single junction when the superconducting coherence length is larger than the dimension of the central superconductor, so that the ABSs present in the system are sensitive to all three superconducting phases. Given that in the experimental work the structure symmetry was also broken, we argue that the physical origin of the diode effect in that system is the same as that discussed here. This is further confirmed by virtually the same critical current dependencies on the biasing phase (which correspond to the perpendicular field in the experimental paper) presented here in Fig. 12 and, most importantly, in the proof-of-concept model Fig. 4, which explicitly realizes the theoretical scenario for the diode effect. Moreover, very recently another experiment demonstrated the presence of the diode effect in phase-biased multi-terminal junctions [31], this time reporting the effect in a *single*, multi-terminal Josephson junction but with the theoretical explanation exploring the tunnel limit [53].

In summary, we demonstrated theoretically the conditions for the appearance of SDE in multi-terminal Josephson junctions, focusing on an example of a three-terminal junction. We showed that SDE is an inherent property of such systems, provided that there is phase bias difference on a pair of superconducting contacts and the presence of at least two current-carrying ABSs, which are characterized by different capabilities to transport the quasiparticles between the superconducting contacts. Our theoretical considerations are compatible with recent experimental findings and confirm the appearance of the diode effect in junctions in both short and long regimes without the need for the presence of spin-related phenomena. Our work paves the way for the realization of nanoscopic Josephson diodes on hybrid multi-terminal structures such as proximitized nanowire crosses [54] or scalable 2DEG platforms [50, 51, 55, 56].

**Funding information**    This work was supported by the program "Excellence Initiative —
research university" for AGH University of Krakow.

**Code availability**    The code used to obtain the results presented in this paper is available in
the on-line repository Ref. [57]

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
