# Peer review of "Theory of universal diode effect in three-terminal Josephson junctions"

_SciPost Physics_

## Round 1 · Referee Report · Anonymous (Referee 1) · 2024-3-31

Report

The authors present a theoretical analysis of a three-terminal Josephson junction and evaluate critical current as a function of phase difference. They show with a simple theoretical analysis that a three-terminal Josephson junction with two ABS can realize JDE due to the relative phase shift between the ABS. I find the analysis and presented simulations for long and short junction limits to be of sound quality and agree with the conclusions. I think the paper can be published in this journal.

I have some minor comments and questions: 1. On page 6 in the description of Figure 4, I am assuming critical current as a function of $\phi_3$ is evaluated by maximizing in $\phi_2$ for a given $\phi_3$, I think it will be helpful to state this clearly. 2. To realize the proposed experimental setup, one would need to apply a magnetic field to phase bias a pair of terminals which breaks the time reversal symmetry. Can authors provide some comments on how their analysis would differ if the time reversal symmetry is broken? 3. In the absence of a phase loop, can authors provide some insight if the diode effect can still be realized in a three-terminal Josephson junction? 4. would like to point out a recent paper on three-terminal Josephson junction where JDE is observed along with $\pi$-supercurrent (https://arxiv.org/abs/2312.17703). Can authors comment if their model can account for this observation?

  • validity: good
  • significance: ok
  • originality: ok
  • clarity: high
  • formatting: good
  • grammar: good

Author:  Jorge Huamani Correa  on 2024-05-27  [id 4517]

(in reply to Report 1 on 2024-03-31)

Thanks to the Referee for the positive comments. We attach a file with the answers and comments about his/her questions.

Attachment:

First_Report.pdf

---

## Round 1 · Referee Report · Anonymous (Referee 2) · 2024-4-8

Strengths

1 - The paper is nicely written and technically correct.
2 - Authors provide both analytical and numerical solutions that supports the claims in the paper and the paper contains a link to the code repository necessary for reproducing the results in the paper.

Weaknesses

1 - The idea of using a trijunction for a superconducting diode effect has been explored before.
2 - The paper does not provide a discussion on which regime (short- or long-junction regime) is better for implementing a superconducting diode.
3 - The paper lacks comparison with the previous three-terminal Josephson diodes reported in the literature.

Report

The manuscript explores the superconducting diode effect (SDE) in multi-terminal Josepshon junction, focusing on a three-terminal setup. Traditionally, the SDE is associated with differences in critical currents for currents flowing in opposite directions in a two-terminal setup. However, the study reveals that in multi-terminal systems, this effect arises naturally without the need for spin interactions, arising instead from a relative shift between Andreev bound states (ABSs) carrying the supercurrent due to phase-biasing the third lead. The analytical findings for the short junction case is supported by numerical simulations and long junction case is investigated through numerical analysis. The results and findings of the paper is technically correct and the paper is nicely written.

On the other hand, the idea of using multi-terminal Josephson junctions to break inversion symmetry and to break time-reversal symmetry by phase-biasing had been explored before. As authors cite in their paper, Ref. [30-31] reports the superconducting diode effect.

The main novelty presented in the current manuscript is the use of a single scattering region. In my opinion, the difference is only incremental and the paper does not present a breakthrough or open a new pathway in the research area of the superconducting diode effect. Therefore, I believe the paper is more suitable for SciPost Phys. Core, and I recommend publishing there if the authors address the questions and comments.

Requested changes

1 - Can authors comment on the impact of the orbital effect due to the applied magnetic field and effects of disorder in the long junction limit?
2 - How much does quasi-continuum states contribute to the diode effect in comparison to the subgap states?
3 - A comparison between short and long junction regime for the superconducting diode effect would enhance the impact of the paper.

  • validity: high
  • significance: good
  • originality: good
  • clarity: high
  • formatting: excellent
  • grammar: excellent

Author:  Jorge Huamani Correa  on 2024-05-27  [id 4518]

(in reply to Report 2 on 2024-04-08)

Thanks to the Referee for careful assessment of our manuscript. We attach a file with the answers and comments about his/her questions.

Attachment:

Second_Report.pdf

---

## Editorial Decision

resubmitted